# A Deep Neural Network Based Glottal Flow Model for Predicting Fluid-Structure Interactions during Voice Production

**Yang Zhang , Xudong Zheng and Qian Xue \***

Department of Mechanical Engineering, University of Maine, Orono, ME 04469, USA;
yang.zhang@maine.edu (Y.Z.); xudong.zheng@maine.edu (X.Z.)
**\*** Correspondence: qian.xue@maine.edu

**Abstract:** This paper proposes a machine-learning based reduced-order model that can provide fast and accurate prediction of the glottal flow during voice production. The model is based on the Bernoulli equation with a viscous loss term predicted by a deep neural network (DNN) model. The training data of the DNN model is a Navier-Stokes (N-S) equation-based three-dimensional simulation of glottal flows in various glottal shapes generated by a synthetic shape function, which can be obtained by superimposing the instantaneous modal displacements during vibration on the prephonatory geometry of the glottal shape. The input parameters of the DNN model are the geometric and flow parameters extracted from discretized cross sections of the glottal shapes and the output target is the corresponding flow resistance coefficient. With this trained DNN-Bernoulli model, the flow resistance coefficient as well as the flow rate and pressure distribution in any given glottal shape generated by the synthetic shape function can be predicted. The model is further coupled with a finite-element method based solid dynamics solver for simulating fluid-structure interactions (FSI). The prediction performance of the model for both static shape and FSI simulations is evaluated by comparing the solutions to those obtained by the Bernoulli and N-S model. The model shows a good prediction performance in accuracy and efficiency, suggesting a promise for future clinical use.

**Keywords:** glottal flow; machine learning; deep neural network; reduced-order modeling

## 1. Introduction

Voiced sound production in the human larynx is a complex fluid-structure interaction process in which the forced air from the lungs interacts with vocal fold tissues to initiate sustained vibrations that modulate the glottal airflow. The waveform of the glottal flow sets the important acoustic parameters of the sound source. One of the important research goals in voice production is to understand the interaction mechanism between glottal aerodynamics and vocal fold tissue mechanics. The fluid-structure interaction during normal vocal fold vibrations has been well understood with the myoelastic-aerodynamic theory [1]. During each vibration cycle, a mucosal wave travels from the inferior aspect to the superior of the vocal folds. The glottis, which refers to the space between the two vocal folds, forms a convergent shape during vocal fold opening and a divergent shape during vocal fold closing. This alternative convergent-divergent glottal shape generates a temporal pressure asymmetry inside the glottis, which ensures sustained energy transfer from airflow to vocal fold tissues to sustained vibrations. In voice disorders, however, this vibration pattern would be disrupted. Voice disorders are often associated with vocal fold pathologies, such as nodule, cyst, scar, paralysis, and so forth. These pathologies alter the geometry and material properties of vocal fold tissues, resulting in irregular mucosal waves. In these conditions, the glottis during vibrations often exhibits irregular shapes [2]. For example,

the glottal channel is often curved from the inferior to superior due to left-right asymmetry. For another example, a multi-channel configuration is often generated due to partial contact of the two vocal folds, which are supposed to be in full contact at the end of the vibration cycle to fully close the glottis. How these irregular glottal shapes affect fluid-structure interactions and the final voice outcome is not well understood. Such understanding is important to elucidate the fundamental mechanism of irregular vocal fold vibrations associated with vocal fold pathologies. Computer models have been playing an important role in understanding the physics of voice production. The very first computer model of voice production was the two-mass model, which modeled each vocal fold as a system of two coupled, spring-mass dampers [3]. The mathematical formulation of the two-mass model was revolutionary in that it provided a very simple and computationally effective manner for computing vocal fold dynamics. The model has been extensively used for studying the physics of voice production in both normal and pathological conditions [4–10]. A shortcoming of this type of model is the lack of physiological correlation between tissue properties and model system parameters [11,12]. Although some effort has been made to establish these relationships [13,14], direct clinical applications are still difficult because a realistic representation of laryngeal physiology is often required in clinics.

Continuum vocal fold models improved upon the lumped-mass model by being able to incorporate the realistic morphology and material properties of vocal folds, therefore having a great potential for clinical applications. While continuum vocal fold models have been greatly improved from simple 2D configurations and isotropic materials to highly complex 3D subject-specific geometries and anisotropic materials [15–19], their use in simulating vocal pathologies is still very limited due to a lack of an accurate and rapid computation of flow pressures in highly irregular glottal shapes. The Bernoulli equation has been dominantly used for its simplicity but it relies on the assumption of symmetric, inviscid one-dimensional (1D) flow and single-channel glottal shape [20–22]. While this assumption is reasonable for normal vocal fold vibrations, it becomes erroneous in many irregular glottal shapes in which the intraglottal flow can be severely curved and the glottis often presents a multiple channel configuration. The Navier-Stokes (N-S) equation can compute the correct flow pressures in these irregular shapes; however, this approach is computationally extremely expensive and as such, is not suitable for clinical use. A major clinical application of a computational voice simulator would be simulation based surgery management, such as predicting the outcome of surgical interventions, optimizing the surgical adjustments, providing patient-specific solutions based on patient-specific anatomy and needs.

It has been shown that self-sustained oscillation of vocal folds is dominated by a few modes of vibration, even when the motion is abnormal [23–26]. This high predictability of the vibratory pattern of the vocal folds stimulated the use of a machine-learning approach to model glottal flow dynamics based on glottal shapes. In this paper, the model is based on the Bernoulli equation with a viscous loss term predicted by a deep neural network (DNN) model. The DNN model is used for two reasons. First, the viscous loss of glottal flow is mainly affected by its channel shape; yet, the relationship is highly complex and nonlinear. The DNN model is well known for its ability in learning complex relationships using a large amount of data [27], which makes it well suitable for learning the relationship in the current problem. Second, vocal fold vibration patterns are dominated by a few vibration modes; therefore, the shapes of the glottal flow channel are highly predictable. By generating flow solutions using a large amount of combinations of vibration modes, vast data can be provided for training the DNN model. The training data of the DNN model is the N-S equation based three-dimensional simulation of glottal flows in various glottal shapes generated by a synthetic shape function. The input parameters of the DNN model are the geometric and flow parameters extracted from discretized cross sections of the glottal shapes and the output target is the corresponding flow resistance coefficient which determines the viscous loss term of the modified Bernoulli equation. The target value of the flow resistance coefficient is calculated from the modified Bernoulli equation where the values of the flow rate and pressure distribution are obtained from the N-S solution. *K*-fold cross validation [27] is performed to fine tune the architecture and hyperparameters of the

DNN. With this trained DNN-Bernoulli model, the flow resistance coefficient as well as the flow rate and pressure distribution in any given glottal shape generated by the synthetic shape function can be predicted. Furthermore, in order to assess the dynamical prediction performance of the DNN-Bernoulli model, a specific fluid-structure interaction (FSI) case of the glottal flow is studied. First, a continuum-mechanics based vocal fold model is coupled with the Bernoulli model to obtain various shapes extracted from one vibration cycle; then based on these shapes, the DNN model is trained in the same way as in the synthetic shape case. Finally, the Bernoulli model in the coupled FSI solver is replaced by the trained DNN-Bernoulli model to predict the glottal vibration dynamics. The prediction performance of the DNN-Bernoulli model in accuracy and efficiency is demonstrated by comparing with the results obtained by the Bernoulli and N-S model.

The outline of the paper is organized as follows: Formulation of the reduced-order model is elaborated in Section 2.1. Implementation of the DNN model for synthetic glottal shapes is discussed in Section 2.2. Implementation of the DNN-Bernoulli model for FSI simulation of a continuum-mechanics based vocal fold model is presented in Section 2.3. Performance of the DNN-Bernoulli model for synthetic shapes and FSI simulation is evaluated in Sections 3.1 and 3.2, respectively. Finally, the conclusions are summarized in Section 4.

## 2. Materials and Methods

### 2.1. Formulation of the Reduced-Order Model

#### 2.1.1. Schematic of the Airway in the Larynx

The schematic of the airway in the larynx is illustrated in Figure 1. The view is in the direction of the length of vocal folds. The airway is composed of three parts, including a contraction part formed by the subglottal part of the vocal folds, the glottis formed by the medial surface of the vocal folds and an expansion part, which is the vocal tract [3]. The vocal folds are modelled discretely with uniformly distributed cross sections along the airflow direction. The definition of the notations in the figure are listed in Table 1.

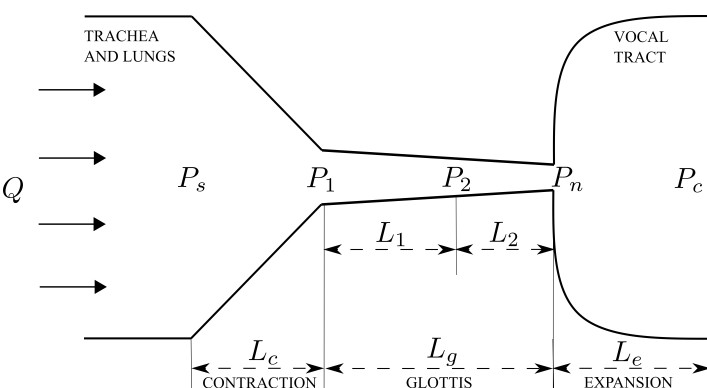

**Figure 1.** Schematic of the vocal folds model.

**Table 1.** Definition of the notations in Figure 1.

| Notation | Definition |
|---|---|
| $Q$ | flow rate |
| $P_s$ | prescribed pressure at the entry of the contraction part |
| $P_1$ | pressure at the exit of the contraction part |
| $P_2$ | pressure at one of the sections of the glottis part |
| $P_n$ | pressure at the exit of the glottis part |
| $P_c$ | prescribed pressure at the expansion part |
| $L_c, L_g, L_e$ | length of the contraction, glottis and expansion parts, respectively |
| $L_1, L_2$ | distance between consecutive sections of the glottis part |

### 2.1.2. Modified Bernoulli Equation

Ishizaka et al. [3] did an early work by adding viscous terms in the Bernoulli equation to predict the pressure pattern inside the glottis. Concretely, in the contraction part, based on the measurements of Van den Berg et al. [28], a loss factor of 0.37 was included to account for the abrupt contraction; in the glottis part, a viscous loss term was added to modify the pressure drop in the Bernoulli equation; in the expansion part, the pressure recovery was estimated with Newton's law. Motivated by their work, we applied a viscous loss term to each section of the glottis and combine them with the Bernoulli equation to obtain the pressure pattern inside the glottis.

Refer to Figure 1, for the contraction and glottis parts, assuming that the vocal folds are discretized with $n + 1$ uniformly distributed cross sections along the airflow direction, then similar to the method in Reference [3], the modified Bernoulli equation which includes the viscous loss can be expressed in the following form:

$$P_s + \frac{\rho}{2}(\frac{Q}{A_s})^2 = P_n + \frac{\rho}{2}(\frac{Q}{A_n})^2 + \frac{\rho}{2}Q^2 \sum_{i=1}^{n} \frac{f_{r_i}}{A_i^2}, \tag{1}$$

where $\rho$ is the air density, $A_i$ is the glottal area at the $i$th cross section, $f_r$ is the flow resistance coefficient of each section of the glottis whose value is not known a priori and needs to be predicted by the DNN model. Details about the implementation of the DNN model will be discussed in Section 2.2.

For the expansion part, similar to the treatment in Reference [3], we have:

$$P_n - P_c = -\frac{\rho}{2}(\frac{Q}{A_n})^2 \cdot 2\frac{A_n}{A_c}(1 - \frac{A_n}{A_c}). \tag{2}$$

Therefore, the pressure at each cross section of the glottis can be written as follows:

$$P_i = P_s + \frac{\rho}{2}(\frac{Q}{A_s})^2 - \frac{\rho}{2}(\frac{Q}{A_i})^2 - \frac{\rho Q^2}{2} \sum_{j=1}^{i} \frac{f_{r_j}}{A_j^2}, \tag{3}$$

where $1 \leq i \leq n$.

### 2.1.3. Calculation of the Flow Rate

In order to calculate the pressure distribution in the glottis from Equation (3), besides the prediction of the flow resistance coefficient $f_r$, the value of the flow rate $Q$ is also required. In previous reduced-order modeling of the vocal folds [20–22], the flow rate was obtained by assuming that the pressure at the minimum sectional area equals to zero and not including the viscous effects. In this subsection, we propose a more accurate way of predicting the flow rate based on the modified Bernoulli model.

Let the index $i = n$ in Equation (3); we have:

$$P_n = P_s + \frac{\rho}{2}(\frac{Q}{A_s})^2 - \frac{\rho}{2}(\frac{Q}{A_n})^2 - \frac{\rho Q^2}{2} \sum_{j=1}^{n} \frac{f_{r_j}}{A_j^2}. \tag{4}$$

By substituting Equation (4) into Equation (2), we can easily derive the expression of the flow rate $Q$ as follows:

$$Q = \sqrt{\frac{2(P_s - P_c)}{\rho\left[ -\frac{1}{A_s^2} + \frac{1 - 2\frac{A_n}{A_c}(1 - \frac{A_n}{A_c})}{A_n^2} + \sum_{j=1}^{n} \frac{f_{r_j}}{A_j^2} \right]}}. \tag{5}$$

Once the value of $f_r$ on each section of the glottis has been predicted by the DNN model, the flow rate $Q$ can be directly obtained from Equation (5).

### 2.2. Implementation of the DNN Model

As discussed in Section 2.1, an accurate prediction of the flow resistance coefficient $f_r$ is the key to the success of the proposed flow model. Assuming that the value of $f_r$ of each cross section of the glottis can be determined by the corresponding local geometrical and physical features, then a fully connected DNN model [27] can be used to establish the mapping relationship between the input features and corresponding output value of $f_r$. In this work, the training data set is the Navier-Stokes equation based three-dimensional simulations of glottal flow in various glottal shapes generated by a synthetic shape function. Given a good selection of geometrical and physical input features and the corresponding target value of $f_r$, the DNN model can be trained. With this trained model, the flow rate in Equation (5) and pressure distribution in Equation (3) in any given glottal shape generated by the synthetic shape function can be predicted in an efficient and accurate way. The workflow of the training and prediction process is illustrated in Figure 2.

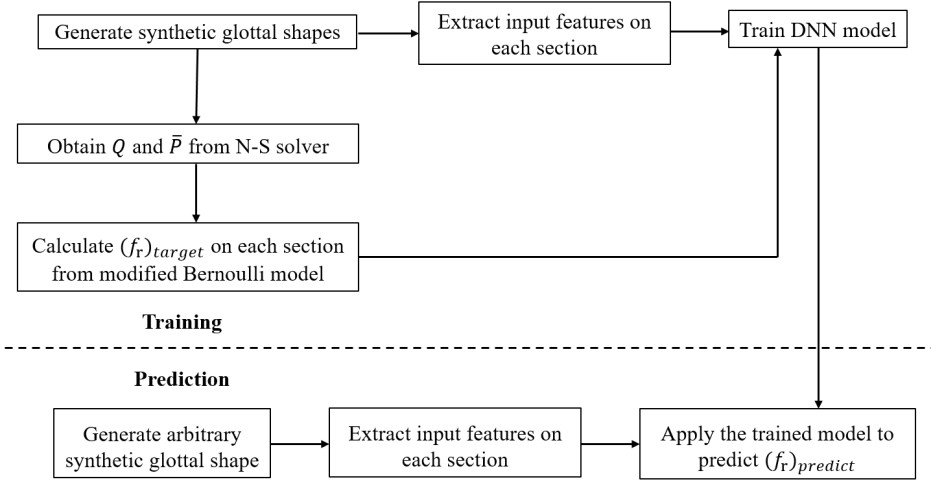

**Figure 2.** Workflow of training and prediction.

### 2.2.1. Synthetic Shape Generation

The synthetic glottal shapes will be generated based on the surface-wave approach [4]. The shape of the medial surface of vocal fold $g(y, z, t)$ at any time instant $t$ during vibration can be obtained by superimposing the corresponding modal displacements $\xi(y, z, t)$ on the prephonatory (initial) geometry $\xi_0(y, z)$, that is,

$$g(y, z, t) = \xi_0(y, z) + \xi(y, z, t), \tag{6}$$

where $y$ and $z$ indicate the inferior-superior (airflow) and anterior-posterior directions, respectively. By specifying different parameter values in the expression of $\xi_0(y, z)$, one can generate convergent or divergent prephonatory shapes [2]. In the expression of modal displacements $\xi(y, z, t)$, the modes of vibration are described with $(m, n)$, where $m$ and $n$ correspond to the number of half-wavelengths in the anterior-posterior and inferior-superior directions, respectively. For details of the expression of $\xi_0(y, z)$, $\xi(y, z, t)$ and the explanation of the corresponding parameters, please refer to Reference [2].

For the purpose of simulating normal vocal fold vibration, we select two different prephonatory shapes, that is, convergent and divergent, the two most dominant modes, that is, $(m, n) = (1, 0), (1, 1)$ and 16 phases during one vibration cycle to generate 64 shapes in total. Note that the contact surface is calculated as an average of the left and right surface coordinates and the contraction and expansion parts are extruded based on the generated medial surface. The resolution of the uniform surface mesh is 0.01 cm. The representative convergent and divergent generated shapes at fully closed, fully open and in between fully closed and fully open phases are illustrated in Figures 3 and 4, respectively.

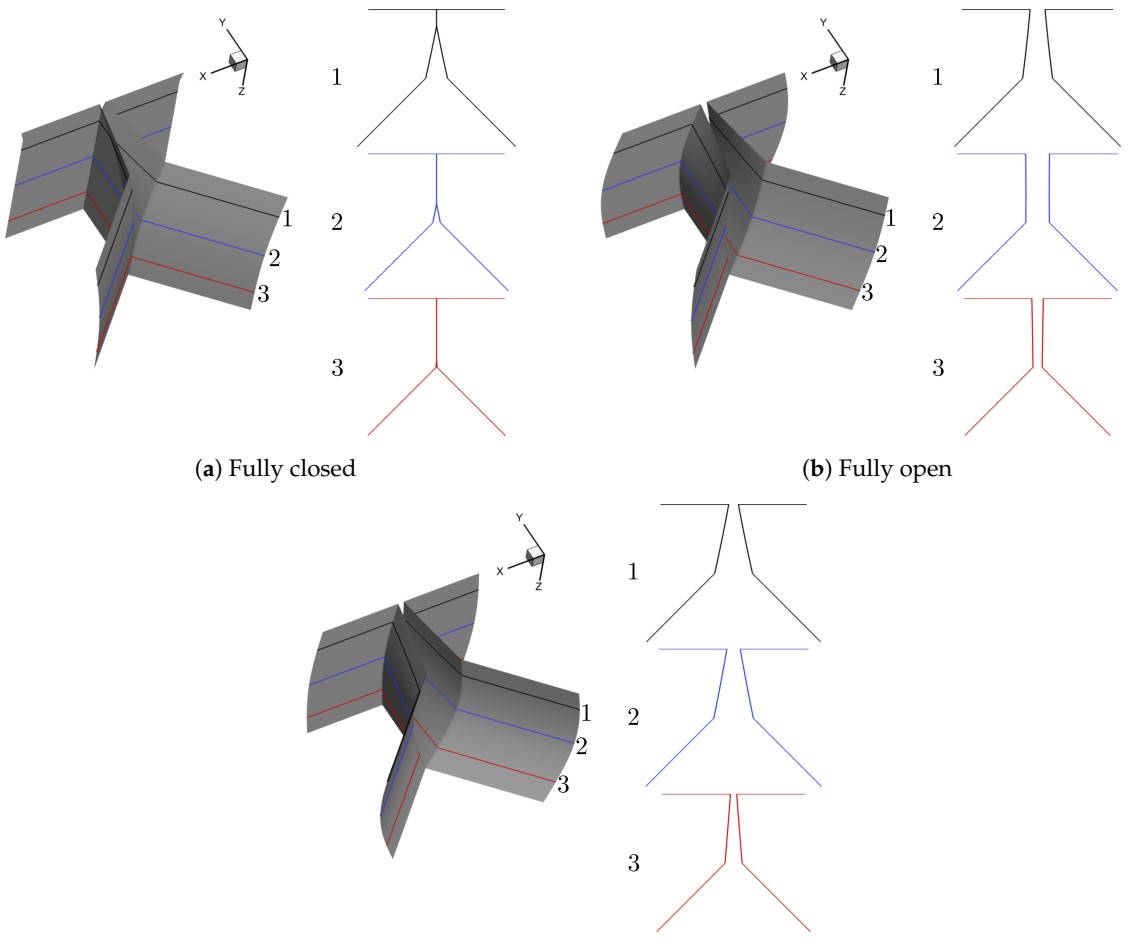

(**a**) Fully closed

(**b**) Fully open

(**c**) In between fully closed and fully open.

**Figure 3.** Representative convergent shape (For each representative shape, three planar surfaces in the $x - y$ plane are extracted along the anterior-posterior ($z$) direction to give a detailed view of the glottis shape and the number of each planar surface is denoted as 1, 2, 3 from the anterior to the posterior direction, respectively).

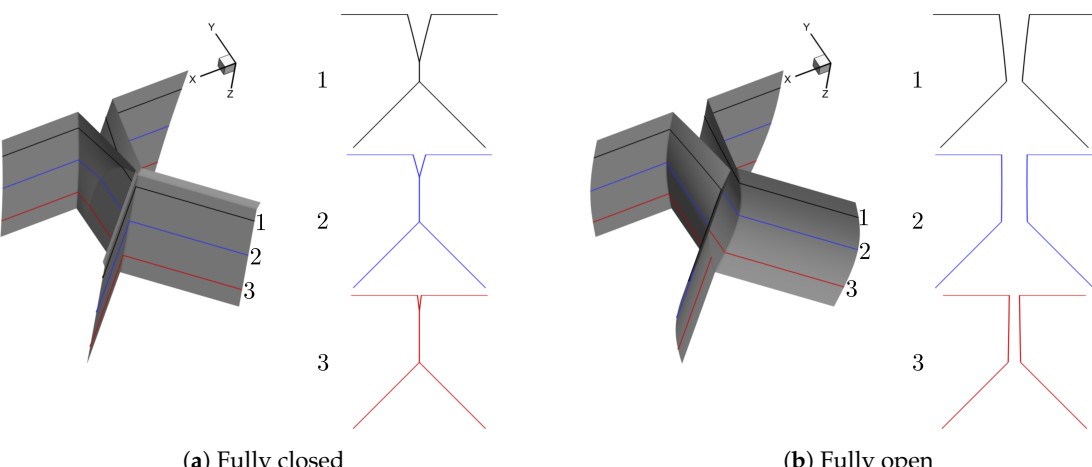

(**a**) Fully closed

(**b**) Fully open

**Figure 4.** *Cont.*

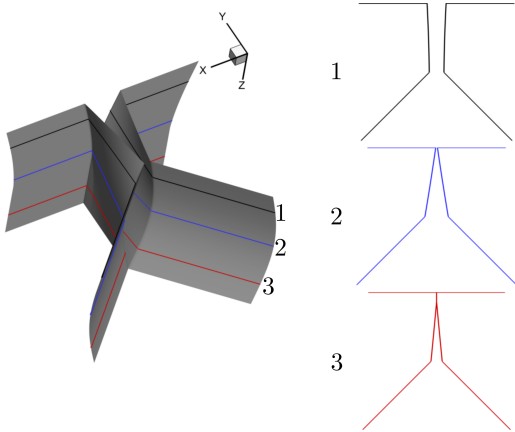

(**c**) In between fully closed and fully open

**Figure 4.** Representative divergent shape (For each representative shape, three planar surfaces in the $x - y$ plane are extracted along the anterior-posterior ($z$) direction to give a detailed view of the glottis shape and the number of each planar surface is denoted as 1, 2, 3 from the anterior to the posterior direction, respectively).

### 2.2.2. Feature Extraction and Target Value of $f_r$

For each of the above 64 shapes, an in-house sharp-interface immersed-boundary N-S flow solver [18] is used to obtain the ground truth values of the flow rate and average pressure on each section along the inferior-superior direction of the vocal folds, then the target value of $f_r$ on each section can be calculated with the following equation:

$$(f_{r_i})_{target} = \frac{(\bar{P}_{i-1})_{NS} - (\bar{P}_i)_{NS} + \frac{\rho}{2}\left[(\frac{Q_{NS}}{A_{i-1}})^2 - (\frac{Q_{NS}}{A_i})^2\right]}{\frac{\rho}{2}(\frac{Q_{NS}}{A_i})^2}, \tag{7}$$

where the subscript $NS$ represents the N-S solution.

For training the DNN model, the vocal folds are uniformly discretized into $n = 128$ cross sections along the inferior-superior ($y$) direction such that the spacing resolution between every two consecutive cross sections is 0.012 cm. As illustrated in Figure 5, a total of 8 local geometrical and physical features at each discretized cross section are defined and extracted, that is, position ($Y_i^*$), area ($A_i^*$), hydraulic diameter ($D_i^*$), upstream angle ($\alpha_i^+$), downstream angle ($\alpha_i^-$), shape change rate ($\Delta S_i^*$), pressure drop ($\Delta P_i^*$) and Reynolds number ($Re_i$) of the $i$ th cross section. The non-dimensional form of those input features are used and the corresponding expressions are listed in Table 2, where $D_i = 4A_i/Pe_i$ is the hydraulic diameter with $Pe_i$ the wetted perimeter [29] of the $i$th cross section, $\Delta P = P_s - P_c$ the total pressure drop along the inferior-superior direction of the vocal folds, $\nu$ the kinematic viscosity coefficient. In this case, the inlet and outlet pressure are set to be $P_s = 1.0$ kPa and $P_c = 0.0$ kPa, respectively and air density and kinematic viscosity are chosen as $\rho = 1.145 \times 10^{-3}$ g/cm$^3$ and $\nu = 1.655 \times 10^{-1}$ cm$^2$/s, respectively.

The input features can be organized as a two-dimensional matrix:

$$x = \begin{bmatrix} Y_1^* & A_1^* & D_1^* & \alpha_1^+ & \alpha_1^- & \Delta S_1^* & \Delta P_1^* & Re_1 \\ Y_2^* & A_2^* & D_2^* & \alpha_2^+ & \alpha_2^- & \Delta S_2^* & \Delta P_2^* & Re_2 \\ \vdots & \vdots & \vdots & \vdots & \vdots & \vdots & \vdots & \vdots \\ \vdots & \vdots & \vdots & \vdots & \vdots & \vdots & \vdots & \vdots \\ Y_m^* & A_m^* & D_m^* & \alpha_m^+ & \alpha_m^- & \Delta S_m^* & \Delta P_m^* & Re_m \end{bmatrix}_{m \times 8}. \tag{8}$$

Similarly, the output target can be written as a vector:

$$\boldsymbol{y} = \begin{bmatrix} fr_1 \\ fr_2 \\ \vdots \\ \vdots \\ fr_m \end{bmatrix}_{m \times 1}, \tag{9}$$

where $m$ is the number of cross sections over all generated shapes.

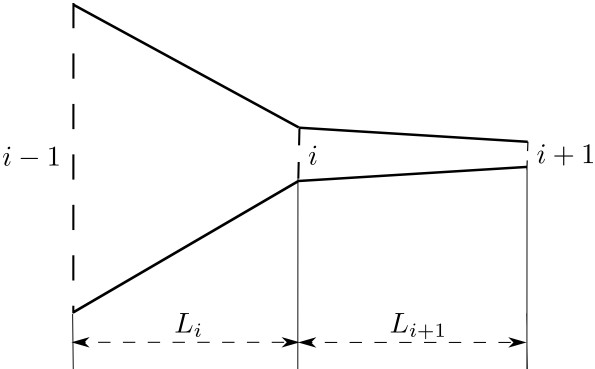

**Figure 5.** Schematic of discretized cross sections.

**Table 2.** Extracted input features.

| Input Features | Non-Dimensional Expression |
| --- | --- |
| Position | $Y_i^* = \frac{Y_i - Y_0}{Y_{n-1} - Y_0}$ |
| Area | $A_i^* = \frac{A_i}{A_0}$ |
| Hydraulic diameter | $D_i^* = \frac{D_i}{D_0}$ |
| Upstream angle | $\alpha_i^+ = \frac{D_i - D_{i-1}}{L_i}$ |
| Downstream angle | $\alpha_i^- = \frac{D_{i+1} - D_i}{L_{i+1}}$ |
| Shape change rate | $\Delta S_i^* = \frac{L_0 \Delta A_i}{L_i A_0}$ |
| Pressure drop | $\Delta P_i^* = \frac{\Delta P}{\frac{\rho}{2}(\frac{Q}{A_i})^2}$ |
| Re | $Re_i = \frac{\frac{Q}{A_i} D_i}{\nu}$ |

### 2.2.3. Implementation of the DNN

The mapping relationship between the input features $\boldsymbol{x}$ and output target $\boldsymbol{y}$ can be established by a fully connected DNN [27,30]. Neurons in the fully connected layer have connections to all neurons of the previous layer,

$$\boldsymbol{y} = \sigma(\boldsymbol{w}\boldsymbol{x} + \boldsymbol{b}), \tag{10}$$

where $\boldsymbol{w}$ is the learnable weights, $\boldsymbol{b}$ is the additive bias and $\sigma$ is the nonlinear activation function.

The loss function $J$ of the DNN is

$$J = \frac{1}{N} \sum_m \|\boldsymbol{y} - \hat{\boldsymbol{y}}\|_2^2 + \lambda \|\boldsymbol{w}\|_2, \tag{11}$$

where $\hat{\boldsymbol{y}}$ is the predicted value and $\lambda$ is the regularization coefficient to prevent the overfitting of the DNN model.

The whole data set $(x, y)$ is randomly split into the training and test sets. Since the training set is relatively small, to avoid the overfitting of the model, we use *k*-fold cross validation [27] to fine tune the architecture and hyperparameters of the DNN, such as the number of hidden layers, the number of neurons on each hidden layer, the initialization of the weights, the activation function, the optimization method, the mini-batch size, the number of epochs and the dropout rate [27]. By convention, the value of *k* is chosen between 5 and 10, therefore the value herein is set to be $k = 5$. The final architecture and hyperparameters of the DNN are chosen from those that have the lowest errors on the validation set. The final DNN model is then trained on the full training set and the prediction performance of the trained model is evaluated on the test set.

The final architecture of the DNN is illustrated in Figure 6. The input layer has 8 neurons which correspond to the input features in Table 2. To increase the training accuracy and efficiency, the input features have been normalized by the zero-mean normalization [27]. The output layer has a single neuron which corresponds to the ground truth value of the flow resistance coefficient $f_r$. The linear activation function is used on the output layer. Besides the input layer and output layer, there are four hidden layers with 256, 64, 16 and 4 neurons on each of them, respectively. To prevent overfitting, each hidden layer is followed with a dropout layer with a 20% dropout rate [31]. All of the weights on each layer are initialized with a random normal distribution. The Rectified Linear Units (ReLU) activation function [32] is used on the hidden layers. The DNN model is optimized using a mean-squared loss function with an adaptive version of the stochastic gradient descent algorithm called Nadam (Nesterov Adam) [33]. To balance between the robustness of stochastic gradient descent and the efficiency of batch gradient descent, mini-batch gradient descent is used [33]. The DNN model is trained with 20,000 epochs and the mini-batch size is 256 for each epoch. The DNN model is implemented on the open-source machine learning platform Keras [34] using TensorFlow [35] as the backend.

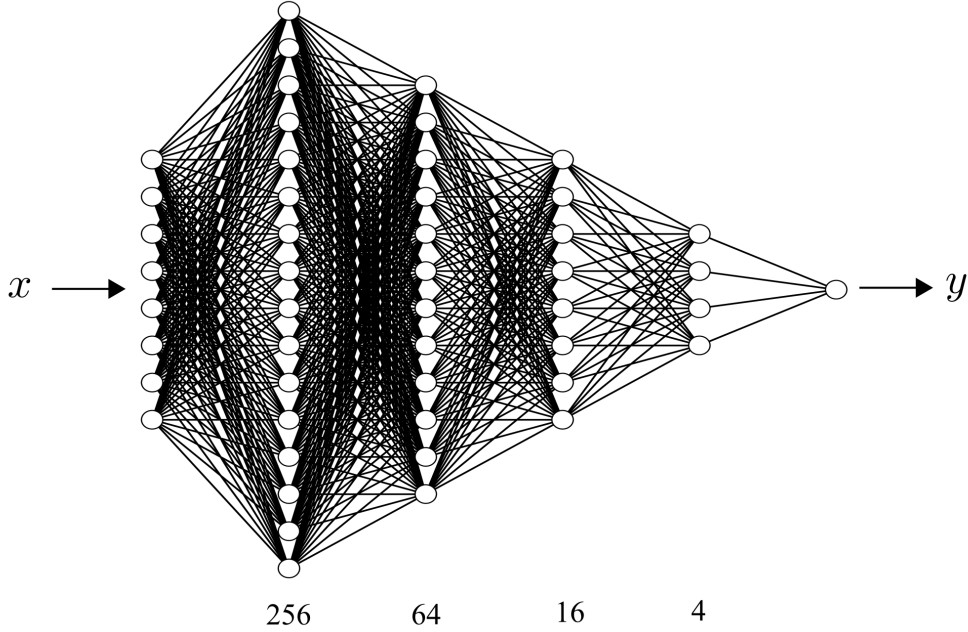

**Figure 6.** Architecture of the deep neural network (DNN).

## 2.3. DNN-Bernoulli Model for FSI Simulation

In the previous subsection, the DNN-Bernoulli model has been implemented for various synthetic shapes generated by Equation (6). In this subsection, we aim to apply the DNN-Bernoulli model in FSI simulation. To this end, FSI simulation is first performed by coupling the Bernoulli model with a finite-element method (FEM) based solid dynamics solver [36] to obtain the self-sustained vibrations. Then various glottal shapes at different time instants of one vibration cycle are extracted and simulated

with the N-S solver. Similar to the procedures in Section 2.2, the input features $x$ are extracted from these shapes, the corresponding output target $y$ is obtained from the N-S solution, the data set $(x, y)$ is fed into the neural network to train and evaluate the DNN model. The final FSI simulation is conducted by coupling the DNN-Bernoulli model with the FEM solver and the results are compared with those obtained by the Bernoulli and N-S solver to demonstrate the improvement of the present model. The abstract workflow is illustrated in Figure 7.

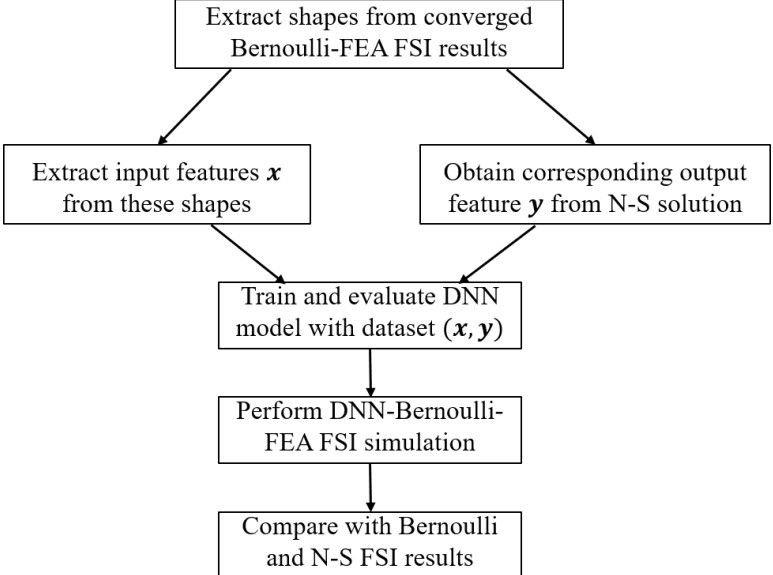

**Figure 7.** Workflow of the DNN-Bernoulli model for fluid structure interaction (FSI) simulation.

Prephonatory Geometry of the Vocal Folds

The prephonatory geometry of a vocal fold (left half) is shown in Figure 8. The length $L$ along the anterior-posterior direction ($z$) and thickness $T$ along the inferior-superior direction ($y$) are 1.5 cm and 0.3 cm, respectively. To simplify the model, only the lateral vibration is allowed, the vertical motion is fixed. An initial gap $\Delta x = 0.002$ cm along the lateral direction ($x$) exists between the left and right counterpart.

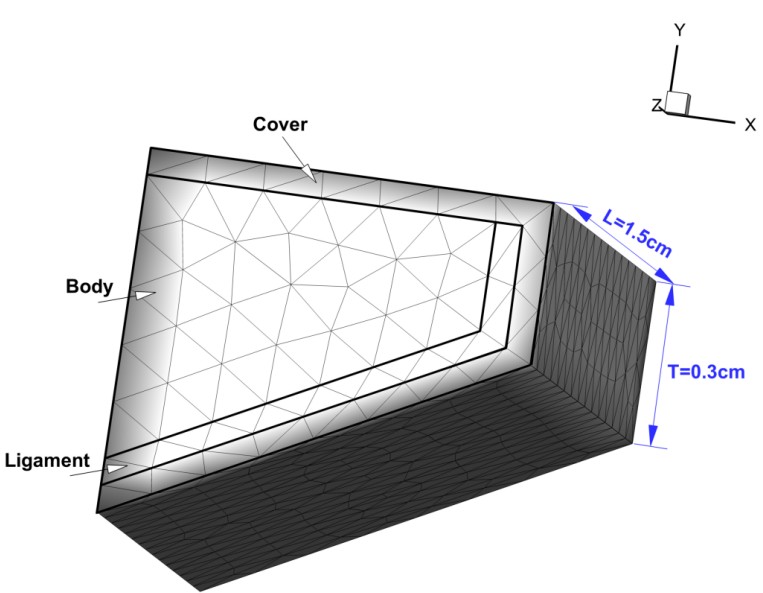

**Figure 8.** Prephonatory geometry of the vocal fold (left half).

Each vocal fold is divided into three layers including the cover, ligament and body. Each layer is assumed to be invariant in the anterior-posterior direction. The vocal fold tissue is modeled as viscoelastic, transversely isotropic material. The material properties of each layer of the vocal fold [15,37] are listed in Table 3.

**Table 3.** Material properties of each layer of the vocal fold.

| | $\rho$ (g/cm$^3$) | $E_p$ (kPa) | $\nu_p$ | $E_{pz}$ (kPa) | $\nu_{pz}$ | $G_{pz}$ (kPa) |
|---|---|---|---|---|---|---|
| Cover | 1.043 | 2.01 | 0.9 | 40 | 0.0 | 10 |
| Ligament | 1.043 | 3.31 | 0.9 | 66 | 0.0 | 40 |
| Body | 1.043 | 3.99 | 0.9 | 80 | 0.0 | 20 |

$\rho$ is the tissue density; $E_p$ and $E_{pz}$ are the transversal and longitudinal Young's Modulus, respectively; $\nu_p$ and $\nu_{pz}$ are the in-plane transversal and longitudinal Poisson ratio, respectively; $G_{pz}$ is the longitudinal shear modulus. [15,37].

## 3. Results and Discussion

### 3.1. Performance of the DNN-Bernoulli Model for Synthetic Shapes

The history of 5-fold cross validation results for synthetic shapes is plotted in Figure 9. The horizontal axis corresponds to the number of epochs and the vertical axis corresponds to the mean absolute error (MAE) between the predicted and target value of $y$. Since dropout is activated when training but deactivated when evaluating on the validation set, MAE on the validation set is smaller than on the training set. The scatter plot of the performance of the trained model on the test set is illustrated in Figure 10. The horizontal and vertical axes correspond to the true and predicted values of $y$ on the test set, respectively. MAE on the training set, validation set and test set are 0.0769, 0.0705 and 0.0487, respectively. The relative MAE divided by the mean ground-truth value of $y$ on the test set is around 6.5%.

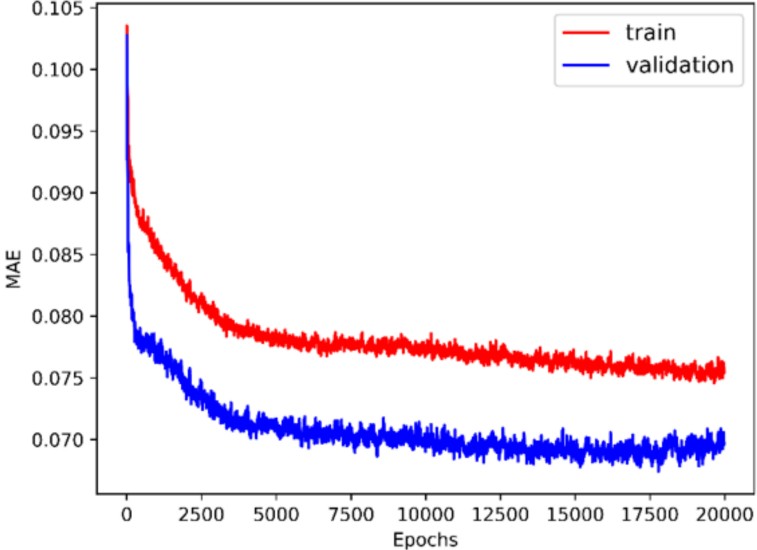

**Figure 9.** Five-fold cross validation results (synthetic shape).

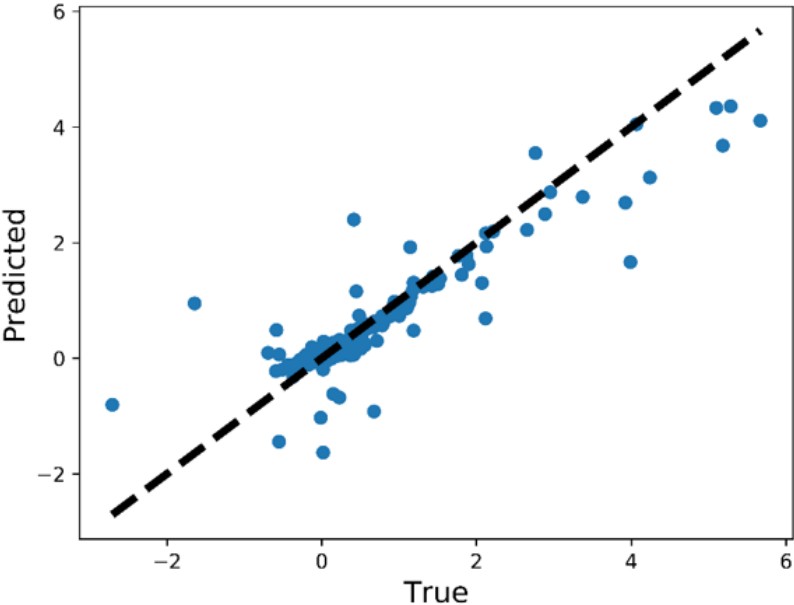

**Figure 10.** Performance of the trained model on the test set (synthetic shape).

To demonstrate the improvement of the present DNN-Bernoulli model over the original Bernoulli equation, we define the relative error of flow rate and pressure distribution on a certain shape as follows:

$$E_Q = \left| \frac{Q - \hat{Q}}{Q} \right| \times 100\% \tag{12}$$

$$E_P = \frac{1}{n} \frac{\sum_{i=1}^{n} \left| P_i - \hat{P}_i \right|}{P_s} \times 100\% \tag{13}$$

where $Q$ and $P_i$ are the flow rate and pressure distribution obtained by the N-S solver, respectively, while $\hat{Q}$ and $\hat{P}_i$ are the corresponding values obtained by the DNN-Bernoulli or Bernoulli model. The comparison of the error range between the DNN-Bernoulli and Bernoulli model on the test set is listed in Table 4. We can see that the error range of both flow rate and pressure distribution of the DNN-Bernoulli model is significantly lower than that of the original Bernoulli model.

**Table 4.** Comparison of error range between Bernoulli and DNN-Bernoulli prediction.

|  | $E_Q$ | $E_P$ |
| --- | --- | --- |
| Bernoulli | 0.27–48.7 | 0.2–19.16 |
| DNN-Bernoulli | 0.01–8.94 | 0.01–8.53 |

Figures 11–14 show the contour of the surface pressure on the vocal folds in three representative glottal shapes predicted by the DNN-Bernoulli, N-S and Bernoulli models. A common phenomenon can be observed from all these figures, that is, the pressure distribution predicted by the DNN-Bernoulli model is much closer to that obtained by the N-S solver, which shows the improvement of the present DNN-Bernoulli model. Note that compared with the Bernoulli model, the additional CPU time required for the present DNN-Bernoulli model during prediction is almost negligible. For the prediction of a newly generated synthetic glottal shape, the average CPU time required for the DNN-Bernoulli model is just around 1 s on a single CPU, while that required for the N-S model is around 2 h per CPU on a parallel computer with 32 CPUs.

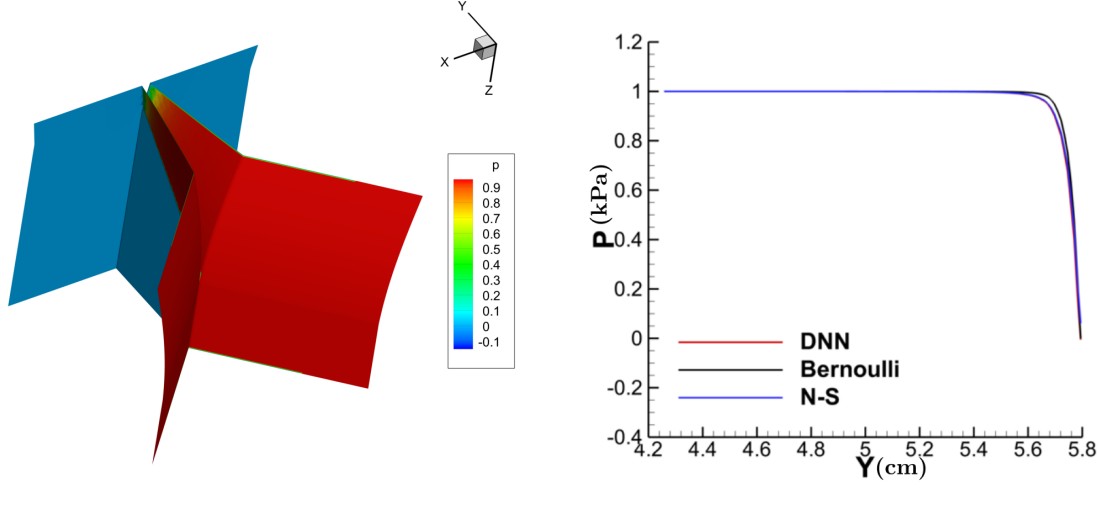

(**a**) Surface pressure contour                    (**b**) Pressure distribution

**Figure 11.** Surface pressure contour and distribution (convergent).

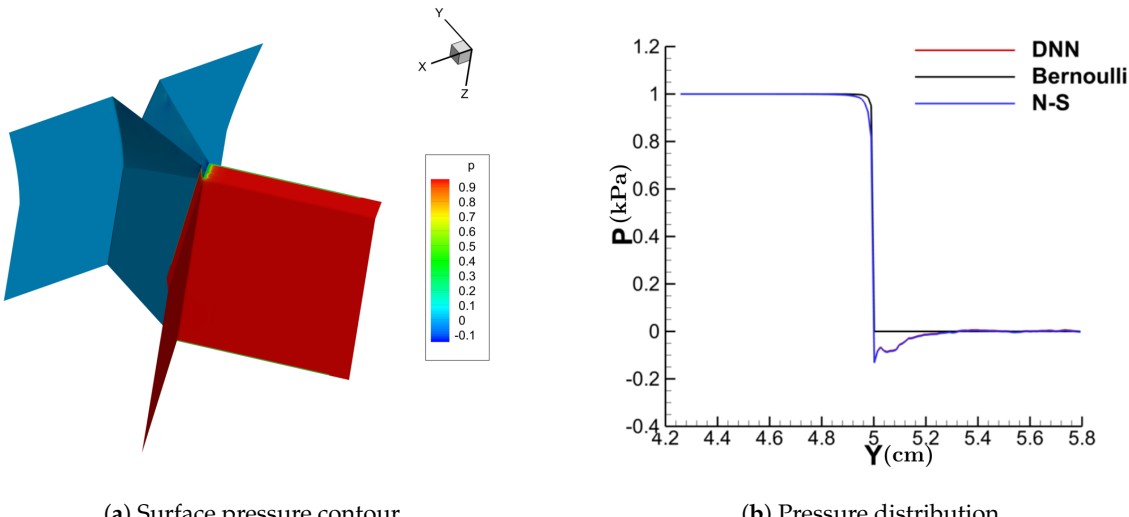

(**a**) Surface pressure contour                    (**b**) Pressure distribution

**Figure 12.** Surface pressure contour and distribution (divergent).

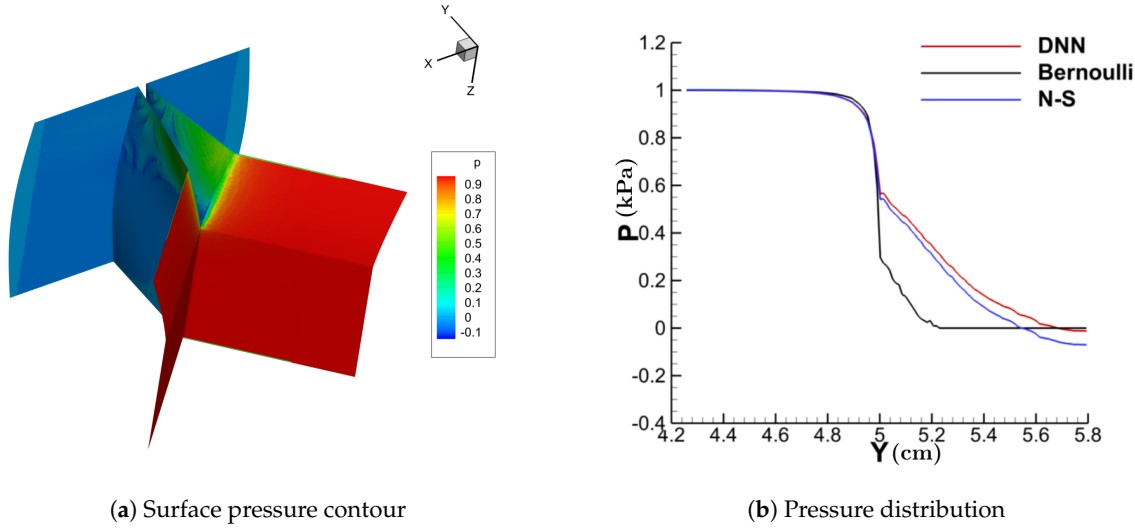

(**a**) Surface pressure contour                    (**b**) Pressure distribution

**Figure 13.** Surface pressure contour and distribution (convergent-divergent).

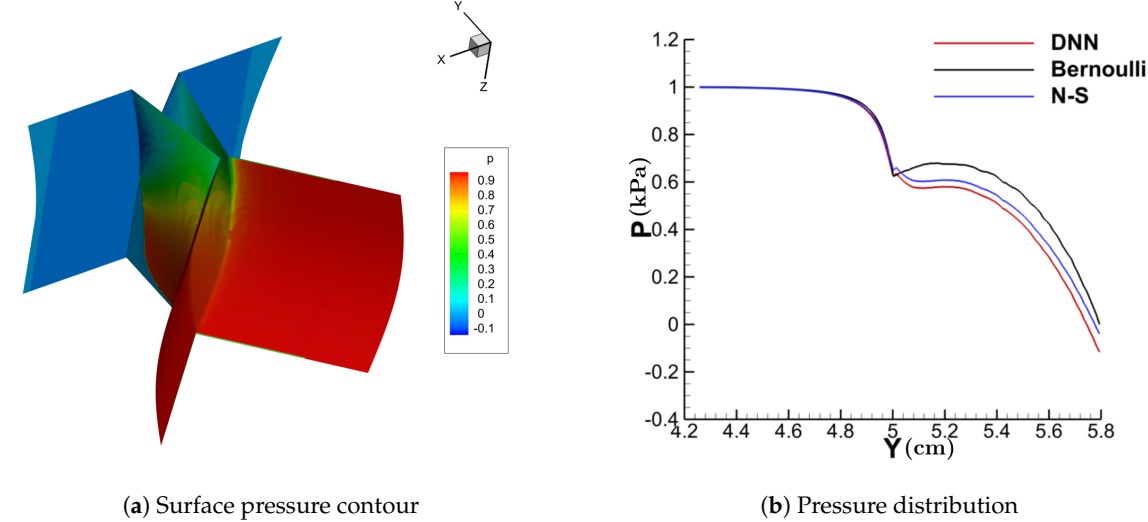

(**a**) Surface pressure contour

(**b**) Pressure distribution

**Figure 14.** Surface pressure contour and distribution (divergent-convergent).

### 3.2. Performance of the DNN-Bernoulli Model for FSI Simulation

Various glottal shapes are extracted from the converged Bernoulli-FEM FSI results within one vibration cycle. The number of extracted shapes is 92 in this case. The input features $x$ listed in Table 2 are extracted from these shapes. The corresponding output target $y$, that is, the target value of $f_r$ is obtained from the N-S solution by Equation (7). The whole data set $(x, y)$ is fed into the same neural network as described in Section 2.2.3 to train and evaluate the DNN model.

The history of 5-fold cross validation results is plotted in Figure 15. The scatter plot of the performance of the trained model on the test set is illustrated in Figure 16. MAE on the training set, validation set and test set are 0.0620, 0.0551 and 0.0393, respectively. The relative MAE divided by the mean ground-truth value of $y$ on the test set is around 4.0%.

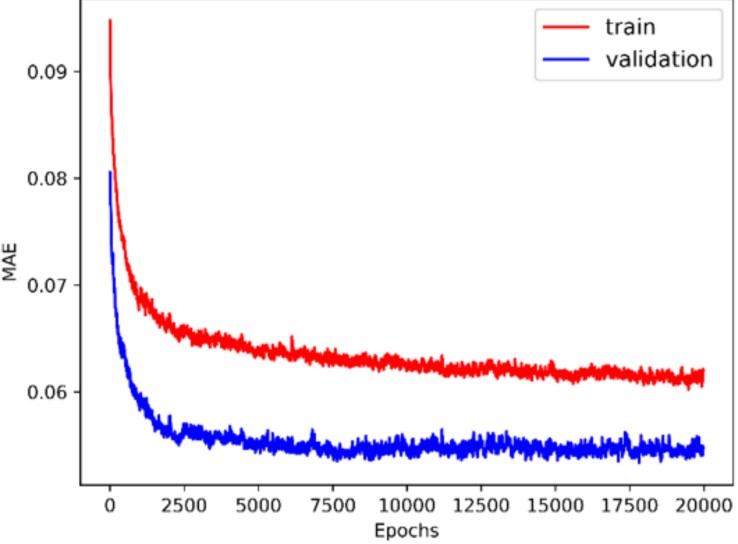

**Figure 15.** Five-fold cross validation results (FSI shape).

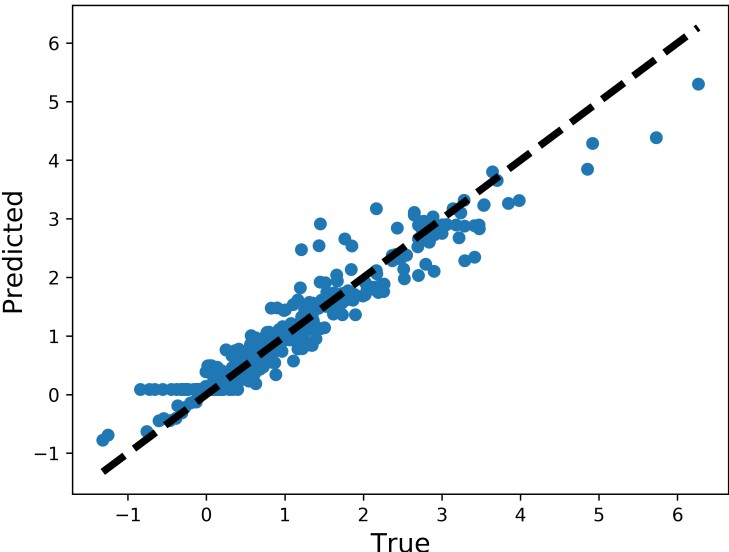

**Figure 16.** Performance of the trained model on the test set (FSI shape).

With the trained model, the DNN-Bernoulli model is coupled with the FEM solver to conduct FSI simulation of the vocal fold vibration. Time history of the flow rate predicted by the DNN-Bernoulli model is compared with that obtained by the Bernoulli model in Figure 17. It can be observed that the maximum value of flow rate is reduced from around 235 mL/s to 135 mL/s with the present model due to the inclusion of the viscous loss. To evaluate if the DNN-Bernoulli model provides a more accurate prediction on the pressure pattern inside the glottis, we extracted the pressure patterns in two representative convergent and divergent glottal shapes from the DNN-Bernoulli results and compared that to those obtained by the Bernoulli and N-S models using the same glottal shapes. The results are illustrated in Figures 18 and 19, respectively. From these figures, we can see that the pressure distribution along the inferior-superior direction of the vocal folds can be well predicted by the present model. Compared with the Bernoulli model, the additional CPU time required for the present DNN-Bernoulli model during prediction is almost negligible.

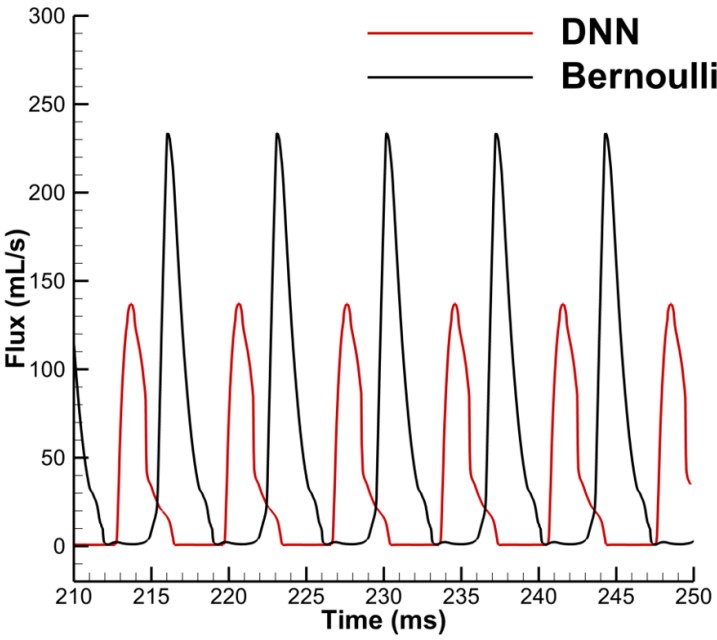

**Figure 17.** Comparison of the flow rate obtained by the DNN-Bernoulli and Bernoulli model.

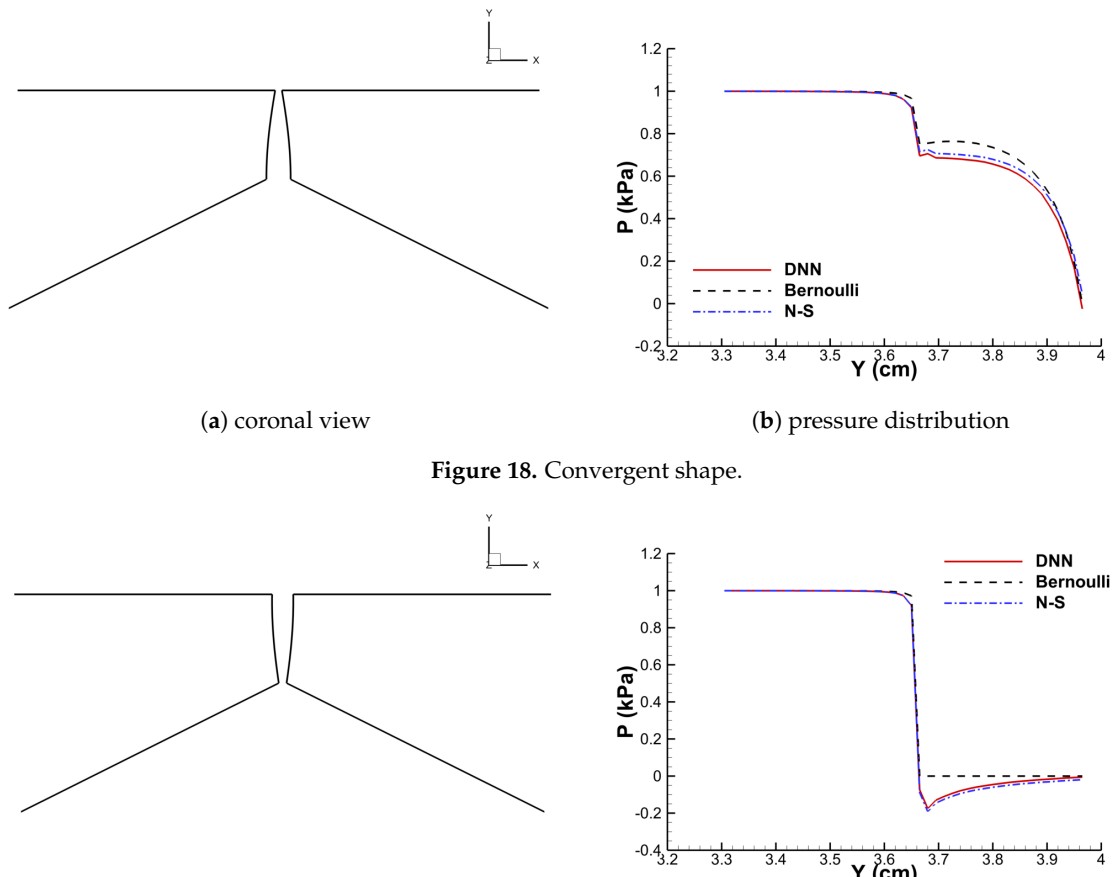

Figure 18. Convergent shape.

Figure 19. Divergent shape.

## 4. Conclusions

A machine-learning based reduced-order model that can provide fast and accurate prediction of the dynamics of the glottal flow is proposed in this paper. The model is based on the Bernoulli equation with a viscous loss term predicted by a DNN model. The training data for the DNN model is collected by generating various glottal shapes using the synthetic shape function. The input features are extracted from discretized cross sections of the generated glottal shapes and the output target is the corresponding flow resistance coefficient that can be obtained from the N-S solution. 5-fold cross validation is performed to fine tune the architecture and hyperparameters of the DNN. With this trained DNN-Bernoulli model, the flow resistance coefficient as well as the flow rate and pressure distribution in any given glottal shape generated by the synthetic shape function can be predicted. Furthermore, a specific FSI case of the glottal flow is studied in order to assess the dynamical prediction performance of the DNN-Bernoulli model. First, a continuum-mechanics based vocal fold model is coupled with the Bernoulli model to obtain various shapes extracted from one converged vibration cycle; then based on these shapes, the DNN model is trained in the same way as in the synthetic shape case. Finally, the Bernoulli model in the coupled FSI solver is replaced by the trained DNN-Bernoulli model to predict the glottal vibration dynamics.

The prediction errors of the DNN model for synthetic shape and FSI cases are around 6.5% and 4.0%, respectively. In terms of the CPU time, compared with the Bernoulli model, the additional CPU time required for the DNN-Bernoulli model is almost negligible once the DNN model is trained. The predicted DNN-Bernoulli results such as the flow rate and pressure distribution for both cases are compared with those obtained by the Bernoulli model and N-S model. The good prediction

performance of the present DNN-Bernoulli model in accuracy and efficiency for both cases shows a great promise for future clinical use.

**Author Contributions:** Conceptualization, Q.X. and X.Z.; methodology, Q.X. and X.Z.; investigation, Y.Z.; writing–original draft preparation, Y.Z.; writing–review and editing, X.Q. and X.Z.; supervision, Q.X. and X.Z.; project administration, Q.X.; funding acquisition, Q.X. All authors have read and agreed to the published version of the manuscript.

**Funding:** This research was funded by the National Institute on Deafness and Other Communication Disorders (NIDCD) grant number 5R21DC016428.

**Acknowledgments:** The project was supported by Grant Number 5R21DC016428 from the National Institute on Deafness and Other Communication Disorders (NIDCD).

**Conflicts of Interest:** The authors declare no conflict of interest.

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
