# Peer review of "A Deep Neural Network Based Glottal Flow Model for Predicting Fluid-Structure Interactions during Voice Production"

_applsci, doi:10.3390/app10020705_

Round 1
Reviewer 1 Report
The manuscript describes the method of fast and accurate prediction of the dynamics of the glottal flow based on the Bernoulli equation combined with DNN model. It is very interesting concept that DNN model trained by elaborate simulations build up the simplified model performance. I dearly hope to see the practical application with more complex vocal fold models.
P.2 L.39: What kind of application does the “clinical use” mean specifically?
P.2 L 44: Please make it clear why did the authors adopt DNN models?
P.6 L.136: The DNN model takes 8 local geometrical and physical features as inputs and predict the flow resistance coefficient. The model is trained by the ground truth values of 64 synthetic glottal shapes. Were the parameters(shapes) for simulating the ground truth values the same in each training epoch, or changed?
P.8 L.158: 5-folds cross validation means that 51 (of 64) glottal shapes were used for training and 13 were used for validation in each fold, is that right? Were there any schemes to assort synthetic data according to prephonatory shapes, vibration modes or phases of vibration cycle in creating the cross validation dataset?
P.13 L.225: What is the reason of the number 92 of extracted shapes?
P.11 L.199, P.13 L.223: The performance of the DNN-Bernoulli model for synthetic shapes and FSI simulation seems to be excellent in the validations. If the geometry of the vocal fold or initial condition is rather different from the training data, how does the model behavior changes? I could not assess whether the model has sufficient generalization ability or robustness because the
P.16 L.260: I desire better understanding of the great advantage of this method that reducing the computational time for simulating glottal flow. How was the difference of computational time between this new method and Navier-Stokes model in practice?
Reviewer 2 Report
This paper examines a machine-learning-based model to provide efficient prediction of the dynamics of speech glottal flow. The model that the authors use is based on the Bernoulli equation, and they use the well-known Navier-Stokes equation. Their model is coupled with a finite-element method-based solid dynamics solver for simulating fluid-structure interactions. The model shows a good prediction performance.
The authors demonstrate solid knowledge of this research area. However, the authors assume that the reader has significant knowledge also of the area, as they use many technical terms without further explanation, as I note below. They assume far too much by the reader, which means that only experts in the area who have worked on the problem that is being discussed can follow what they describe. I give many examples of these problems below.
Specific needed changes:
An Improved Reduced-Order Glottal Flow Model Using Deep Neural Network
->
An Improved Reduced-Order Glottal Flow Model Using a Deep Neural Network
“… the dynamics of the glottal flow.” - say that this is speech; “the glottal flow” presumes too much
… is the Navier-Stokes (N-S) equation based …
->
… is a Navier-Stokes (N-S) equation-based …
… shapes generated by a synthetic shape function. - what is a “synthetic shape function”?
…the output target is the corresponding flow resistance coefficient. - why would this be a useful output?
…distribution along any given glottal shape… - “along”?
… solid dynamics solver … - what is this?
…simulations are evaluated by…
->
…simulations is evaluated by…
… great promise for future clinical use.
->
… promise for future clinical use.
…flow was the two-mass model which modeled each
-.
…flow was the two-mass model, which modeled each
… material properties of vocal fold, therefore
… material properties of vocal folds, therefore
While continuum vocal fold model has been greatly improved
While continuum vocal fold models have greatly improved
…[13–17], its use in
…[13–17], their use in
… in highly irregular glottal shapes. - why are such relevant?
…its simplicity but, it relies on
->
…its simplicity, but it relies on
… as spindle shape, hourglass shape and severely asymmetric shape, … - how are these defined? and why are they relevant?
… glottis often presents a multiple channel configuration. - why? how?
…is computational extremely expensive
…is computationally extremely expensive
…oscillation of vocal folds are dominated
…oscillation of vocal folds is dominated
…of the vibratory pattern the vocal folds
…of the vibratory pattern of the vocal folds
inspired the use … - inspired whom?
of machine-learning approach to
of a machine-learning approach to
…is the corresponding flow resistance coefficient. - it is perhaps not needed to explain this in the abstract, but is surely need here. what is this? why relevant?
5-fold cross validation is … - why 5-fold?
… vibration cycle, then based on
… vibration cycle; then based on
…simulation are evaluated in Subsection 3.1 and 3.2,
…simulation is evaluated in Subsections 3.1 and 3.2,
… expansion part which is vocal tract.
… expansion part, which is the vocal tract.
…in the contraction part,… - what is this part exactly? say more about the geometry of the glottal area here
a loss factor of 0.37 … - what is this? how determined? why relevant?
… for the abrupt contraction; … - what is this?
… in the expansion part, … - explain this more too
Illuminated by their work, …
Motivated by their work, …
we applied viscous loss term to
we applied a viscous loss term to
Ps prescribed pressure at the entry… - why are some of these “prescribed”?
… pressure at certain section… - “certain”?
of the glottis part - what does this mean?
… length of the contraction, glottis and expansion part, respectively
… lengths of the contraction, glottis and expansion parts, respectively
… assuming that the glottal is
… assuming that the glottal section is
what is the reference source for eq. 1?
… at the ith cross section, - - do not use italics for “-th”, only for the “i”
the glottic can
the glottis can
… of vocal folds [18–20],
… of the vocal folds [18–20],
Let the index i = n in Eq. (3), we have:
Let the index i = n in Eq. (3); we have:
each section of the glottal has been predicted by the DNN model,
each section of the glottis has been predicted by the DNN model,
…is Navier-Stokes
…is the Navier-Stokes
…on the prephonatory geometry… - what does “prephonatory” mean?
…get convergent or divergent prephonatory shapes. - what are these?
…number of half-wavelengths … - of what relevance are these?
… is 0.01cm.
… is 0.01 cm.
in Figure 3 and 4, respectively.
in Figures 3 and 4, respectively.
…are extracted along the anterior-posterior direction… - “extracted along”?
Figures 3-4 need more explanation; what do the 1,2,3 refer to? there no labels on any of the axes
…each section of the glottal, … the text repeatedly, and wrongly, uses this word as a noun
…hydraulic diameter,upstream and downstream angles, shape change rate, pressure drop, and Reynolds number (Re). - none of these is explained, or sourced
wetted perimeter … - what is this? (A serious flaw of this paper is its frequent use of undefined technical terms)
… 0.012cm, and the number of cross sections for each shape is n = 128.
… 0.012 cm, and the number of cross sections for each shape is n = 128.
(also, always put a space after a digit here, and before the unit)
When specific numbers are used, as here, it is best to justify the choices
the training and test set.
the training and test sets.
… from those who have the lowest errors
… from those that have the lowest errors
…features have been normalized. - explain more
…layers with 256, 64, 16, and 4 neurons… - why these choices?
… 20% dropout rate [29]. - yet another specific fact in the text, with no context or explanation (there are many of these in this paper; the authors know what they mean and their significance, but they do not state such for the readers here)
… trained with 20000 epochs and the mini-batch size is 256 for each epoch. - why these choices?
… as the backend. - what is this?
…o obtain the self-oscillations. -what are these?
Then various glottal shapes during one vibration cycle are extracted … -in what sense are these “extracted”?
…of the Vocal Fold
…of the Vocal Folds
… of the vocal fold (left half)
… of a vocal fold (left half)
The vocal fold is divided
Each vocal fold is divided
Young’s Modulus, … longitudinal Poisson ratio, … longitudinal shear - what are all these? why relevant?
…corresponds to the number of epochs,… - yet another technical term that is undefined in this text, assuming that the reader is fully aware of all these issues
The horizontal and vertical axis
->
The horizontal and vertical axes
… model on the test set are
… model on the test set is
The common phenomenon can be observed from all these figures, i.e.,
A common phenomenon can be observed from all these figures, i.e.,
The prediction error of the DNN model for
The prediction errors of the DNN model for
References: Capitalize the first letters of all names of sources, e.g.,
Bell system technical journal -> Bell system technical journal
Journal of sound and vibration
Journal of Sound and Vibration
Round 2
Reviewer 2 Report
I appreciate the thorough and detailed revisions that were made based on my earlier comments. I have only the following minor changes to make:
Voiced sound production in human larynx is a complex fluid-structure interaction process
->
Voiced sound production in the human larynx is a complex fluid-structure interaction process
which modulate the glottal airflow. -> that modulate the glottal airflow. The waveform of glottal flow sets the important acoustic … -> The waveform of the glottal flow sets the important acoustic … (I am certain what “sets” means here…)
One of important research goals in voice production is to understand ->
One of the important research goals in voice production is to understand vibrations have been well understood by the … vibrations has been well understood with the … The glottis, which is referred to the space between The glottis, which refers to the space between …. of the two vocal folds which are supposed to be in full contact at …. of the two vocal folds, which are supposed to be in full contact at … interactions and final voice outcome is not well understood. … interactions and the final voice outcome is not well understood. is important for elucidate the fundamental mechanism of irregular is important to elucidate the fundamental mechanism of irregular The vocal folds are discretized with uniformly The vocal folds are modelled discretely with uniformly Referred to Figure 1, for the contraction and glottis parts, …. Refer to Figure 1, for the contraction and glottis parts, …. … DNN, such as the the number of hidden layers, the … DNN, such as the number of hidden layers, the
